# Mitochondrial Respiration of Platelets: Comparison of Isolation Methods

**DOI:** 10.3390/biomedicines9121859

**Published:** 2021-12-08

**Authors:** Andrea Vernerova, Luiz Felipe Garcia-Souza, Ondrej Soucek, Milan Kostal, Vit Rehacek, Lenka Kujovska Krcmova, Erich Gnaiger, Ondrej Sobotka

**Affiliations:** 1Department of Analytical Chemistry, Faculty of Pharmacy in Hradec Kralove, Charles University, Akademika Heyrovskeho 1203/8, 500 05 Hradec Kralove, Czech Republic; verneran@faf.cuni.cz (A.V.); lenkakrcmova@seznam.cz (L.K.K.); 2Department of Clinical Biochemistry and Diagnostics, University Hospital Hradec Kralove, Sokolska 581, 500 05 Hradec Kralove, Czech Republic; 3Oroboros Instruments GmbH, Schoepfstrasse 18, A-6020 Innsbruck, Austria; luiz.garcia@oroboros.at (L.F.G.-S.); erich.gnaiger@oroboros.at (E.G.); 4Department of Clinical Immunology and Allergology, University Hospital Hradec Kralove, Sokolska 581, 500 05 Hradec Kralove, Czech Republic; ondrej.soucek@fnhk.cz; 54th Department of Internal Medicine—Hematology, University Hospital Hradec Kralove, Sokolska 581, 500 05 Hradec Kralove, Czech Republic; kostal.milan@fnhk.cz; 6Transfusion Department, University Hospital Hradec Kralove, Sokolska 581, 500 05 Hradec Kralove, Czech Republic; vit.rehacek@fnhk.cz; 7D.Swarovski Research Laboratory, Department of General and Transplant Surgery, Medical University of Innsbruck, Christoph-Probst-Platz 1, Innrain 52, A-6020 Innsbruck, Austria; 83rd Department of Internal Medicine—Metabolic Care and Gerontology, University Hospital Hradec Kralove, Sokolska 581, 500 05 Hradec Kralove, Czech Republic

**Keywords:** mitochondria, human platelets, thrombocytes, density gradient centrifugation, platelet apheresis, flow cytometry, high-resolution respirometry, oxidative phosphorylation

## Abstract

Multiple non-aggregatory functions of human platelets (PLT) are widely acknowledged, yet their functional examination is limited mainly due to a lack of standardized isolation and analytic methods. Platelet apheresis (PA) is an established clinical method for PLT isolation aiming at the treatment of bleeding diathesis in severe thrombocytopenia. On the other hand, density gradient centrifugation (DC) is an isolation method applied in research for the analysis of the mitochondrial metabolic profile of oxidative phosphorylation (OXPHOS) in PLT obtained from small samples of human blood. We studied PLT obtained from 29 healthy donors by high-resolution respirometry for comparison of PA and DC isolates. ROUTINE respiration and electron transfer capacity of living PLT isolated by PA were significantly higher than in the DC group, whereas plasma membrane permeabilization resulted in a 57% decrease of succinate oxidation in PA compared to DC. These differences were eliminated after washing the PA platelets with phosphate buffer containing 10 mmol·L^−1^ ethylene glycol-bis (2-aminoethyl ether)-*N*,*N*,*N*′,*N*′-tetra-acetic acid, suggesting that several components, particularly Ca^2+^ and fuel substrates, were carried over into the respiratory assay from the serum in PA. A simple washing step was sufficient to enable functional mitochondrial analysis in subsamples obtained from PA. The combination of the standard clinical PA isolation procedure with PLT quality control and routine mitochondrial OXPHOS diagnostics meets an acute clinical demand in biomedical research of patients suffering from thrombocytopenia and metabolic diseases.

## 1. Introduction

Platelets (thrombocytes, PLT) are subcellular elements of blood and substantially contribute to hemostasis by clumping and initiating the formation of blood clots [1]. PLT are fragments of larger cells located in the bone marrow, called megakaryocytes, which are released into the peripheral blood. Despite being absent of nucleus, PLT contain other organelles, such as mitochondria and endoplasmic reticulum.

Mitochondria are key cellular organelles responsible for the production of adenosine triphosphate (ATP), redox homeostasis, regulation of reactive oxygen species and intracellular calcium concentration, activation of apoptosis, and many other functions. Fundamental mitochondrial functions can be studied by assessment of the rate of oxygen consumption related to substrate oxidation and coupling control [2,3]. Supported by high-resolution respirometry (HRR) [4], the analysis of PLT respiration got into the forefront of research interest in biomedical fields [5,6,7,8,9,10,11]. Mitochondrial dysfunction of PLT was observed in several human physiological and pathological conditions, including type II diabetes [12,13], aging [14,15], asthma [16], sepsis [8,17,18,19,20], schizophrenia, Huntington’s, Parkinson’s, and Alzheimer’s diseases [13,21].

Transfusions are used for various medical conditions to replace lost components of the blood. Transfusions of human PLT are necessary for the treatment of pathological conditions associated with low PLT count (thrombocytopenia) or their dysfunction (thrombocytopathy). These transfusions have been used worldwide in clinical practice for decades. The therapeutic benefits of PLT transfusions are generally acknowledged by medical professionals. The process of separating PLT by platelet apheresis (PA) and storing PLT concentrates is well documented and based on many years of good clinical outcomes [22,23,24]. However, improvement of the lifespan and function of stored PLT is still under investigation [25,26].

Isolation of PLT using density gradient centrifugation (DC) is the experimental approach in biomedical research [5]. However, different types of anticoagulants, centrifugation setups, and respiratory protocols affect the behavior of PLT concentrates and the rate of mitochondrial respiration [27,28]. Evaluation and standardization of these procedures are the aims of the European COST project MitoEAGLE (COST Action CA15203) [5]. Mitochondrial respiration of PLT obtained by PA has not yet been evaluated nor compared with PLT isolation by DC.

In view of the potential significance for a broad clinical practice, in the present study, we compared mitochondrial respiration of PLT isolated by PA and DC from the blood of healthy donors.

## 2. Materials and Methods

### 2.1. Subjects

Twenty-nine healthy blood donors registered in a database of the Transfusion Department of University Hospital Hradec Kralove were included in this study. All attendees fulfilled the conditions of health and safety criteria for PLT donation. The study was approved by the ethical committee of the University Hospital Hradec Kralove, Czech Republic (Nr. 201903511P). Written informed consent was obtained from each volunteer before enrolment in the study together with a brief questionnaire about their medical and professional history, physical activity, tobacco and alcohol use, etc. Two PLT samples (for DC and PA) were obtained from each participant to eliminate interpersonal variability and measure mitochondrial respiration simultaneously in both preparations. Whole blood was collected for DC isolation before the initiation of single-donor PA. Subsequently, a sample of PLT concentrate was obtained from final transfusion bags.

### 2.2. Reagents

Calcium-free Dulbecco’s phosphate-buffered solution (DPBS) was obtained from Lonza, Switzerland. Ficoll-Paque™, ethylene glycol-bis (2-aminoethyl ether)-*N*,*N*,*N*′,*N*′-tetra-acetic acid (EGTA), pyruvate, oligomycin, carbonyl cyanide 4- (trifluoromethoxy) phenylhydrazone (FCCP), glucose, rotenone, succinate, digitonin, dimethyl sulfoxide (DMSO), cytochrome c and antimycin-A were obtained from Sigma (Sigma-Aldrich, St. Louis, MO, USA). MiR05-Kit was purchased from Oroboros Instruments (Innsbruck, Austria). All antibodies (anti-CD41 phycoerythrin (PE), clone P2; IgG1 (mouse) fluorescein isothiocyanate (FITC); anti-CD62P FITC, clone CLB-Thromb/6; anti-CD63 FITC, clone CLB-Gran/12) were manufactured by Beckman Coulter (Miami, FL, USA).

### 2.3. Blood Sampling

PLT obtained by PA and DC originated from the same set of healthy PLT donors. Blood withdrawals were scheduled before the initiation of PA and were performed by experienced nurses at the Transfusion Department, University Hospital Hradec Kralove. Whole blood samples were collected into three 6 mL dipotassium ethylenediaminetetraacetic acid (K_2_EDTA) tubes and one 2 mL coagulation citrate sodium tube (Vacuette, Greiner Bio-One GmbH, Kremsmünster, Austria). Blood samples were transported at room temperature (RT) and protected from sunlight to the laboratory of the Department of Clinical Biochemistry and Diagnostics (University Hospital in Hradec Kralove, Czech Republic) for the isolation of PLT using DC and mitochondrial respiration measurements. Flow cytometry was performed in the lab of the Department of Clinical Immunology and Allergology (University Hospital in Hradec Kralove, Czech Republic). Whole blood counts and PLT isolated by both methods were counted on the Sysmex cell counter (Sysmex Europe GmbH, Norderstedt, Germany) before and after sample preparation, respectively.

### 2.4. Platelet Preparation

#### 2.4.1. Density Gradient Centrifugation

To obtain a high quality of human PLT, we followed the latest standard operating procedures and recommendations of the MitoEAGLE network [5] (Figure 1). PLT were isolated from whole blood (12 mL), centrifuged using DC in Leucosep^TM^ tubes (50 mL, Greiner Bio-One GmbH, Kremsmünster, Austria) with 15 mL of Ficoll-Paque^TM^. The blood sample was diluted with sterile DPBS (12 mL) and gently poured on top of the polyethylene barrier and centrifuged at 1000 *g* for 10 min at RT using a swinging bucket (first centrifugation: intermediate acceleration 6, brakes 0). Part of the supernatant (5 mL) was collected into a new tube for later use. The peripheral blood mononuclear cells and PLT-enriched layer (buffy coat) were gently collected using a Pasteur pipette and washed with 25 mL DPBS, followed by a 120 *g* centrifugation for 10 min (second centrifugation: RT, acceleration 6, brakes 2). After obtaining the buffy coat, we used differential centrifugation for the following isolation steps. The PLT-rich supernatant was collected and combined with the plasma from the first centrifugation and 10 mmol·L^−1^ EGTA (final concentration). The PLT were centrifuged at 1000 *g* for 10 min (third centrifugation: RT, acceleration 6, brakes 2), and the pellet was resuspended and washed in 5 mL DPBS with 10 mmol·L^−1^ EGTA and centrifuged at 1000 *g* for 5 min (fourth centrifugation: RT, acceleration 6, brakes 2). The final PLT pellet was resuspended with 0.5 mL DPBS containing 10 mmol·L^−1^ EGTA. The entire PLT isolation protocol by DC required about 60 min to be concluded.

#### 2.4.2. Platelets Apheresis from a Single Donor

Two multicomponent blood collection systems, the Haemonetics MCS+ (Boston, MA, USA) and Trima Accel (San Diego, CA, USA) were used for PA. The MCS+ was used for patients No. 5, 7, 9, 11, 15, and 28, and Trima Accel was used for all other patients. The resuspension solution SSP+ PLT additive solution (Macopharma, Tourcoing, France) contains Na_3_-citrate 2∙H_2_O (108.13 mmol·L^−1^), Na-acetate 3∙H_2_O (32.48 mmol·L^−1^), NaH_2_PO_4_ 2·H_2_O (6.73 mmol·L^−1^), Na_2_HPO_4_ (21.48 mmol·L^−1^), KCl (4.96 mmol·L^−1^), MgCl_2_ 6∙H_2_O (1.48 mmol·L^−1^), NaCl (69.30 mmol·L^−1^) in 1000 mL water (pH 7.2). T-PAS+ additive solution (Terumo BCT Europe N.V., Zaventem, Belgium) is distinguished from SSP+ solution just in Na_2_HPO_4_ (54.17 mmol·L^−1^). Blood was collected by standard venipuncture (16 G) and anticoagulated with acid-citrate-dextrose in a 10:1 ratio for TRIMA and a 9:1 ratio for Haemonetics MCS+. TRIMA and Haemonetics MCS+ were centrifuged at 3000 rpm and 5500 rpm, respectively. Approximately 150 mL of PLT including 30–40 mL of plasma and citrate with 300 mL of resuspension solution was obtained from each donation with a total PLT count of around 400 × 10^9^ cells (in two bags). Each bag consists of plasma and resuspension solution (SSP+ or T-PAS+) in the ratio 30:70. For measurements, 8 mL of final PLT-rich plasma concentrates were sampled aseptically using a sterile connecting device (TSCD II, Terumo Europe N.V., Belgium). The samples were stored in highly gas-permeable storage bags at RT without agitation for 1 h to reduce PLT activation according to clinical recommendations [29]. Afterward, PLT concentrates were stored on a flatbed agitator (TB-80 + RS-50, Tool spol. s.r.o., Prague, Czech Republic) at 22 °C ± 2 °C according to the European Committee (Partial Agreement) on Blood Transfusion and the European Commission [29]. The PA isolation procedure required from 60 to 110 min to be concluded.

#### 2.4.3. Apheresis Sample Washing

To differentiate between the effect of isolation media and the procedure itself, apheresis samples were washed with the same dilution medium as in DC. PA subsamples (about 5 to 10 mL; *n* = 17) were collected into a 15 mL Falcon tube, diluted in 5 mL DPBS with 10 mmol·L^−1^ EGTA, and centrifuged at 1000 *g* for 5 min at RT (acceleration 9, brake 2). The supernatant was discarded. The pellet was gently resuspended in 5 mL DPBS with 10 mmol·L^−1^ EGTA and centrifuged at 1000 *g* for 5 min at RT (acceleration 9, brake 2). The final supernatant was discarded. The pellet was gently resuspended in 0.5 mL DPBS with 10 mmol·L^−1^ EGTA, forming the washed apheresis (WA) group.

### 2.5. Cell Counting

Samples were diluted 10 times by DPBS and counted on the Sysmex cell counter XN-10 series (Sysmex Europe GmbH, Norderstedt, Germany) according to the manufacturer’s instructions. The DC PLT yield was calculated as the number of PLT obtained after sample preparation per total number of PLT in the whole blood (prior to sample preparation).

### 2.6. Flow Cytometry

CD62P and CD63 were used as markers for PLT activation and degranulation. Before and after isolation, all samples (blood with K_2_EDTA and citrate sodium, yields of DC, PA, and WA samples) were investigated in the lab of the Department of Clinical Immunology and Allergology (University Hospital in Hradec Kralove, Czech Republic). Afterward, samples were diluted ten times in physiological saline solution. A volume of 25 µL of each diluted sample was added to tubes containing 3.5 µL of fluorochrome-labeled monoclonal antibodies, including anti-CD41 phycoerythrin (PE), clone P2; IgG1 (mouse) fluorescein isothiocyanate (FITC); anti-CD62P FITC, clone CLB-Thromb/6; and anti-CD63 FITC, clone CLB-Gran/12. Three tubes were prepared for each sample, each containing an anti-CD41 antibody for PLT detection and an antibody against the expression parameter (IgG1, CD62P, CD63). Samples were incubated with antibodies for 10 min at RT in the dark. A volume of 750 µL of physiological saline solution was then added to each tube. The Navios 10 flow cytometer (Beckman Coulter, Prague, Czech Republic) and Kaluza C 1.1 Analysis Software (Beckman Coulter, Prague, Czech Republic) were used. The data at a minimum of 50,000 events were obtained for each staining and supplied as a list mode. PLT were gated as CD41+ events. Expression of CD62P and CD63 was determined as a percentage of positive events compared to IgG isotype control and as mean fluorescence intensity (MFI). The Citrate and K_2_EDTA blood samples were used as a control for comparison of activation of PLT. CD62P (P-selectin) is an antigen that shows the rate of PLT activation, while the expression of the CD63 marker serves to detect activated basophils, and the level of expression correlates well with their degranulation.

### 2.7. Mitochondrial Respiration

PLT respiration was measured by HRR using Oroboros O2k-FluoRespirometers (Oroboros Instruments, Innsbruck, Austria) equipped with 0.5 mL Duran^®^ glass chambers containing MiR05 [30] at 37 °C. For calculation of the PLT concentration in the chamber, the dead volume of the stopper capillaries (0.04 mL) was considered, resulting in a total volume of 0.54 mL before closing the chamber at 0.5 mL. For the addition of PLT to the chamber, the partial volume replacement approach was used, i.e., before the addition of PLT, the corresponding volume of MiR05 was removed. 100 · 10^6^ to 120 · 10^6^ PLT [19,31] were added and kept under constant stirring during the substrate-inhibitor-titration (SUIT) protocol. Data were recorded in real-time with DatLab 7.4 software (Oroboros Instruments, Innsbruck, Austria), with a data recording interval of 2 s.

### 2.8. SUIT Protocol for HRR

The SUIT-003 coupling control and cell viability (CCV) protocol [32] was used to study coupling control and plasma membrane permeability of living cells (Figure 2) [33,34]. Respiratory capacities are tested in a sequence of coupling states: ROUTINE respiration *R*, LEAK compensatory respiration *L*, and electron transfer (ET) capacity *E* [34,35].

After stabilization of endogenous ROUTINE respiration (ce1), 10 mmol·L^−1^ of pyruvate (ce1P) was added, followed by titration of oligomycin (ce2Omy; ATP-synthase inhibitor; 5 to 10 nmol·L^−1^) to induce LEAK respiration. The uncoupler FCCP (ce3U) was titrated stepwise to evaluate ET capacity. The Complex I inhibitor rotenone (ce4Rot) was then added to inhibit NADH-linked respiration. Subsequently, the Complex II substrate succinate (10 mmol·L^−1^; ce5S) was added for evaluation of the plasma membrane integrity. The plasma membrane of all cells was then permeabilized with digitonin (1Dig). To access mt-outer membrane integrity, cytochrome *c* was titrated (1c). Finally, the Complex III inhibitor antimycin A (2Ama) was added to fully inhibit mitochondrial respiration for evaluation of residual oxygen consumption (*Rox*).

### 2.9. Statistical Analysis

Data are expressed as median with interquartile range (IQR). Bar graphs with two columns were analyzed with the Wilcoxon matched-pairs signed rank test and bar graphs with three or more columns were analyzed by one-way ANOVA with multiple comparison test (Tukey), multiple t-tests using the Holm–Sidak method, and by 2-way ANOVA using multiple comparisons (Sidak’s multiple comparisons test). Individual *p* values were expressed to four decimal places in each figure. All statistical analyzes were performed using GraphPad Prism version 9.0 (GraphPad Software, San Diego, CA, USA).

## 3. Results

### 3.1. Characteristics of the PLT Donors

Twenty-nine healthy PLT donors (27 males and 2 females) were included in the study with an average age of 38.5 ± 7.3 years, body mass 92 ± 43 kg, and height 184.0 ± 7.2 cm. None of the PLT donors was diagnosed with diabetes, depression, nor suffered serious hematological or other internal comorbidities. The characteristics of the PLT donors are summarized in Appendix A. There was no significant dependence of PLT respiration on age, body mass, or body mass index (BMI) (Appendix A), nor on night shifts and cigarette smoking (Appendix A).

### 3.2. Mitochondrial Respiration of PLT

Representative traces of PLT respiration are shown in Figure 2. Oligomycin-inhibited LEAK respiration did not differ between PLT preparations. PLT isolated by PA had 22% higher ROUTINE respiration (*R*_ce1P_) and 16% higher ET capacity (*E*_ce3U_) in comparison to DC (*p* = 0.001; Figure 3A). In contrast, the S-linked ET capacity of permeabilized cells (S*_E_*_,1Dig_) was lower in the PA group at 57% of DC (*p* < 0.001; Figure 3A). Effective concentrations of oligomycin, FCCP, and digitonin had to be increased 2.2-, 3.5- and 2.5-times, respectively, in the PA group in comparison to DC (Figure 4).

These results suggest that compounds of the donor’s plasma were carried over from PA isolation to the respiration medium, supporting on the one hand (1) higher respiratory rates in the living cells, on the other hand (2) buffering the effects of oligomycin, uncoupler and digitonin, and (3) inhibiting respiration in permeabilized PLT. This hypothesis was tested in the WA group, obtained by washing a part of the PA suspension with DPBS containing 10 mmol·L^−1^ EGTA, which chelates Ca^2+^. Strikingly, all differences in mitochondrial respiration of PLT isolated by DC and PA were diminished in the WA group (Figure 3A). Flux control ratios (*FCR*) using ET capacity *E* as a reference state represent an internal normalization independent of the cell count [3]. No differences were observed in the *R*/*E* flux control ratio between all groups, but the *FCR* for the S-pathway was significantly lower in PA in comparison to DC and WA (Figure 3B). This points to a specific shift in S-pathway capacity when using different isolation protocols. The presence and concentration could impact respiration leading to a decreased S-linked ET capacity.

### 3.3. Yield and Sample Quality: Cytochrome C Test and Viability

Total yields of PLT from whole blood averaged 68% using DC. Because of the methodological principle of PA, it is not possible to calculate the PLT yield from PA. However, after washing the apheresis sample with DPBS and centrifugation at 1000 *g* for five minutes, the resuspended concentrates (i.e., WA) contained a similar concentration and yield of PLT as the samples isolated by DC (Appendix A).

We evaluated the PLT quality using the respiratory viability index (*VI*_R_, Figure 5A) and cytochrome *c* test (Figure 5B). *VI*_R_ was slightly but significantly higher in PLT isolated by DC versus PA, reaching 87% vs. 83% (*p* = 0.026; Figure 5A). The intact plasma membrane of platelets is impermeable to externally added succinate, which therefore can stimulate respiration only in cells with functional mitochondria but a damaged cell membrane, which are considered as dead cells. Permeabilization of the plasma membrane by digitonin titration (protocol step 1Dig) provides a reference state of 100% permeabilized cells with respiration in the entire mitochondrial population supported by succinate in the noncoupled state [34]. The viability index was restored in the WA group to the high level of the DC group (Figure 5A).

The cytochrome *c* test indicated good structural integrity of the outer mitochondrial membrane in all groups (Figure 5B).

### 3.4. Platelet Activation

To investigate the role of PLT activation we assessed the expression of the protein receptor P-selectin (CD62P) and the PLT activation marker CD63 on the PLT plasma membrane. Whole blood collected using citrate sodium (Citrate) and K_2_EDTA (EDTA) tubes, and isolated PLT obtained from DC, PA, and WA were analyzed by flow cytometry from samples of 17 patients (Figure 6).

PLT were not activated during DC and PA blood isolation (Figure 7). Increased ROUTINE respiration in the PA group, therefore, was not due to the increased PLT activation during PA. The results for WA showed a similar trend as those for DC. Taken together, our data show higher signs of PLT activation and mitochondrial respiration of DC and WA samples.

## 4. Discussion

Differential diagnosis of human diseases is a systematic process based on several diagnostic tools and procedures ranging from non-invasive approaches, such as physical examination, ultrasound, or X-ray imaging, to invasive methods, such as endoscopy, exploratory laparoscopy, or tissue biopsy. Laboratory analysis of human blood is a minimally invasive method that can be repeated several times without exposing the patient to severe risks. Examination of human PLT function is a promising topic for hematology, since suitable methods are available, and standardization is achievable. Optical aggregometry is a routine method for the examination of PLT function [36], but the specific clinical interpretation is difficult. Determination of PLT count and size, and PLT visualization by electron microscopy are insufficient for describing PLT function. The PLT count may create a false impression of bleeding diathesis since even a low PLT count of about 20–30×10^9^ x·L^−1^ is sufficient for adequate hemostasis in profound immune thrombocytopenia [37]. Knowledge of functional PLT activity in these cases could help the prophylaxis of anticoagulant treatment and improve the patients’ quality of life. Therefore, analysis of mitochondrial respiration of PLT represents a promising approach [38,39,40,41,42]. Increased respiration of PLT isolated from the blood of septic patients correlates with activation of circulating proinflammatory cytokines [17]. The bioenergetics of PLT is affected in diabetic rats, by increasing PLT respiration, mitochondrial mass, and membrane potential [43].

PA is the gold standard used for many years in the clinical environment to obtain PLT concentrates used for thrombocyte transfusions. The treatment potential of this method is corroborated by years of relevant clinical results. On the other hand, the DC isolation protocol [5] was developed to preserve mitochondrial function for respirometric analysis of PLT isolated from small amounts of blood obtained by venipuncture. The question arises on how PA affects mitochondrial respiration of PLT in comparison to PLT isolated by DC.

In our study, the PA isolation procedure affected the mitochondrial respiratory fingerprint of human PLT. ROUTINE respiration and ET capacity of living PLT was higher in the PA group compared to the DC group (Figure 3A). Respiratory fuel substrates contained in the serum support higher ROUTINE respiration and ET capacity, indicated by the higher ET capacity in living PLT suspended in plasma compared to PBS with glucose [19]. Pyruvate stimulated ROUTINE respiration of living PLT (Figure 2) and the S-pathway ET capacity of permeabilized PLT was higher than the ET capacity of living PLT. These results corroborate that the ET capacity of living PLT is substrate-limited and does not reflect the maximum capacity of mitochondrial electron transfer in PLT [19]. Taken together, additional external substrates provided in PA are primary candidates for stimulation of both ROUTINE respiration and the apparent ET capacity in living PLT.

It is well established that a higher optimum uncoupler concentration for stimulation of maximum respiration (ET capacity) is required in cells incubated in culture media compared to MiR05. In endothelial cells, the optimum FCCP concentration is 6–8 µM in culture medium RPMI and 1–2 µM in mitochondrial respiration medium [44]. To our knowledge, oligomycin titrations have not yet been reported to evaluate the effect of incubation media on the minimum oligomycin concentration required to fully inhibit the ATP synthase and phosphorylation of ADP. This can be explained by the fact that much higher concentrations of oligomycin have been used previously in respiratory studies, which entail, however, inhibition of subsequently measured ET capacity [34].

In contrast to ET capacity in living cells, after permeabilization, the S-linked ET capacity was inhibited by more than 40% in the PA group compared to the DC group. A higher optimum concentration of digitonin for permeabilization of the plasma membrane is required in cells incubated in culture media compared to MiR05 [45]. The candidate compound is Ca^2+^, which stabilizes the plasma membrane and thus protects it from permeabilization at low digitonin concentrations. External Ca^2+^ brought into contact with mitochondria after permeabilization, in turn, inhibits respiration, which provides an explanation for the inhibition of S-linked ET capacity in PA. The inhibitory effect was reversed by washing with DPBS containing 10 mmol·L^−1^ EGTA, a strong chelator of free Ca^2+^ ions. Despite the lack of added Ca^+2^ in the apheresis solution, SSP+ PLT additive solution, residual calcium from the donor’s plasma could still be present in the PA samples, and even though we used two different systems for PA, we did not observe any significant differences between them (data not shown). These results indicated that the choice of medium for storage and the further respirometric measurements might be key considerations when using PLT as a biomarker of mitochondrial changes as suggested by Siewiera et al. [46].

Addressing PLT activation, we performed flow cytometry and assessed the panel of activation markers. Based on preliminary experiments, we measured two membrane antigens CD62P and CD63. PLT isolated by DC and WA showed higher activation in comparison to the PA group (Figure 7). It was reported previously, however, that some degree of spontaneous activation is always related to platelet collection and EDTA causes higher expression of activation markers in comparison to Citrate [47,48]. Taking into consideration that blood for each group (DC and PA) was withdrawn into tubes using different anticoagulant chemicals, EDTA and Citrate respectively, the important information is the change of activation markers in comparison to their respective controls. Neither DC nor WA caused higher PLT activation in comparison to their EDTA control. A similar trend was observed for PA, which did not exert higher activation than its control counterpart (Figure 7). Therefore, we conclude that the isolation process itself did not cause any additional activation of PLT.

Lastly, we investigated the relationship between markers of PLT activation and mitochondrial respiration (Figure 8) by combining data from all groups and performing regression analysis [49]. We did not find an apparent pattern, since all *r*^2^ values were quite low (data not shown). The highest correlation was found for the S pathway, when expressed relative to ET capacity of living cells as *FCR* (*r*^2^ = 0.27; Figure 8A), suggesting a possible trend, especially when we paired results of PLT are paired for each donor (Figure 8B).

Of note, PLT isolated by PA from each donor were less activated and had lower respiration in the S-pathway. Our results, however, do not demonstrate causality between PLT activation and differences in mitochondrial respiration, since we do not provide any data regarding negative and positive controls. This is one of the limitations of our study and to further clarify this phenomenon, more experiments using specific PLT activators and inhibitors are necessary. The second limitation of this study is that we do not provide data on PLT aggregatory functions. Considering that PLT activation to a certain level is still a reversible process, specific measurements must be performed (such as optical aggregometry). In this study, we demonstrated that HRR is a method sensitive enough to study PLT metabolism, but to further investigate PLT metabolism during hemostasis exceeds the framework of this manuscript. The advantage of this study is that we provide respirometric reference values of human PLT isolated by protocol developed in the international project MitoEAGLE [5], allowing our results to be used in comparative mitochondrial physiology in future projects.

## 5. Conclusions

Using PA as the clinical method for PLT isolation preserved mitochondrial integrity and function equally well in comparison to the faster isolation by DC. However, PA compared to DC resulted in (1) stimulation of respiration in living PLT, (2) higher effective concentrations of oligomycin, FCCP, and digitonin, and (3) inhibition of ET capacity after permeabilization. We did not observe any significant effect of PLT isolation methods on PLT activation in comparison to their controls. All observed differences in respiratory control were reversed in the WA group by an additional washing step after PA preparation. The differences between DC and PA, therefore, were caused by compounds carried over from the serum to the respiratory assay in the PA group. In conclusion, subsamples obtained from clinical PA can be used after a simple washing step (WA) for analysis of mitochondrial respiratory control in living and permeabilized PLT. High-resolution respirometry can thus be integrated with standard clinical isolation procedures for PLT quality control and routine diagnostics of mitochondrial respiratory function by OXPHOS analysis.

## Figures and Tables

**Figure 1 biomedicines-09-01859-f001:**
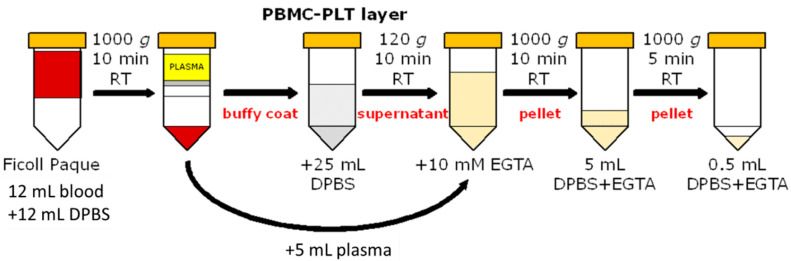
Scheme of platelets (PLT) preparation using density gradient centrifugation (DC). From the first centrifugation of whole blood with Ficoll-Paque^TM^ to the final pellet, which was resuspended in Dulbecco’s phosphate-buffered solution (DPBS) and ethylene glycol-bis (2-aminoethyl ether)-*N*,*N*,*N*′,*N*′-tetra-acetic acid (EGTA) at room temperature (RT). The PLT suspension was counted (10 times diluted) and added to the O2k-chamber at a final cell concentration of 200 × 10^6^ x·mL^−1^. PBMC, peripheral blood mononuclear cell.

**Figure 2 biomedicines-09-01859-f002:**
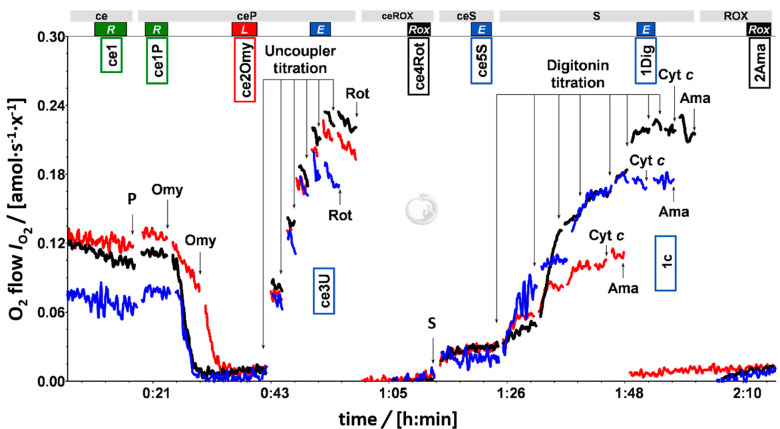
Representative traces of PLT respiration of DC (black), platelet apheresis (PA; red), and washed apheresis (WA; blue) samples. Oxygen flux per cell, *I*_O2_ [amol·s^−1^·x^−1^]. Coupling control and cell viability (CCV) protocol with living cells (ce1), pyruvate addition (ce1P), oligomycin titration (10 to 40 nmol·L^−1^; ce2Omy), uncoupler titration of carbonyl cyanide 4-(trifluoromethoxy)phenylhydrazone (FCCP) at 0.1 µmol·L^−1^ steps (ce3U). Rotenone for residual oxygen consumption (ce4Rot; 1 µmol·L^−1^ ROX). Cell viability test in the CCV protocol: succinate addition (ce5S; 20 mmol·L^−1^) with digitonin titration (1Dig). Cytochrome c test (1c; 10 µmol·L^−1^) and antimycin-A addition for *Rox* (2Ama; 2.5 µmol·L^−1^).

**Figure 3 biomedicines-09-01859-f003:**
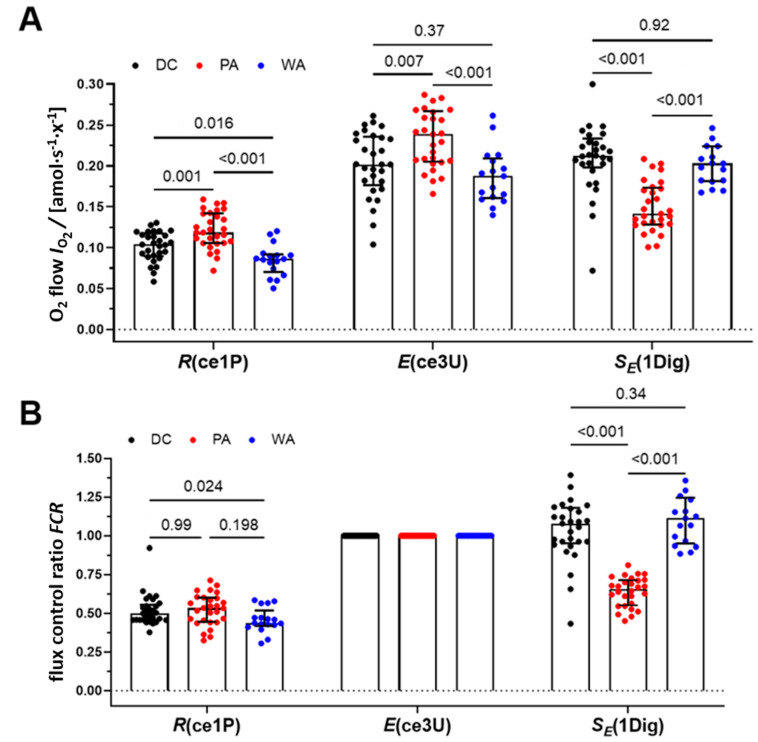
Effect of isolation methods on PLT respiration. PLT isolated by DC (black dots), PA (red dots), and WA (blue dots). (**A**) Oxygen flow per cell, *I*_O2_ [amol·s^−1^·x^−1^], in the ROUTINE- and ET-state of living cells (*R*_ce1P_ and *E*_ce3U_) and S-linked ET capacity of permeabilized cells (S*_E_*_,1Dig_). (**B**) Flux control ratio (*FCR*). *E*_ce3U_ was selected as the reference state. Two-way ANOVA, *p* values are shown above their respective pairwise comparison bar.

**Figure 4 biomedicines-09-01859-f004:**
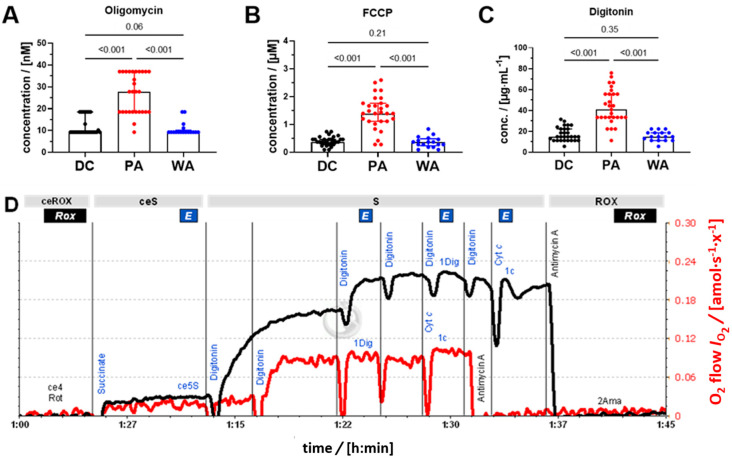
Effect of isolation methods on the inhibitory concentration of oligomycin, and optimum concentrations of uncoupler FCCP and digitonin. (**A**) Adenosine triphosphate (ATP)-synthase inhibitor oligomycin. (**B**) Uncoupler FCCP. (**C**) Mild detergent digitonin for plasma membrane permeabilization. (**D**) Representative trace of the viability module of the SUIT protocol. DC (black) and PA (red). Oxygen flow per cell *I*_O2_ [amol·s^−1^·x^−1^]. One-way ANOVA with multiple comparison test (Tukey), *p* values are shown above their respective pairwise comparison bar.

**Figure 5 biomedicines-09-01859-f005:**
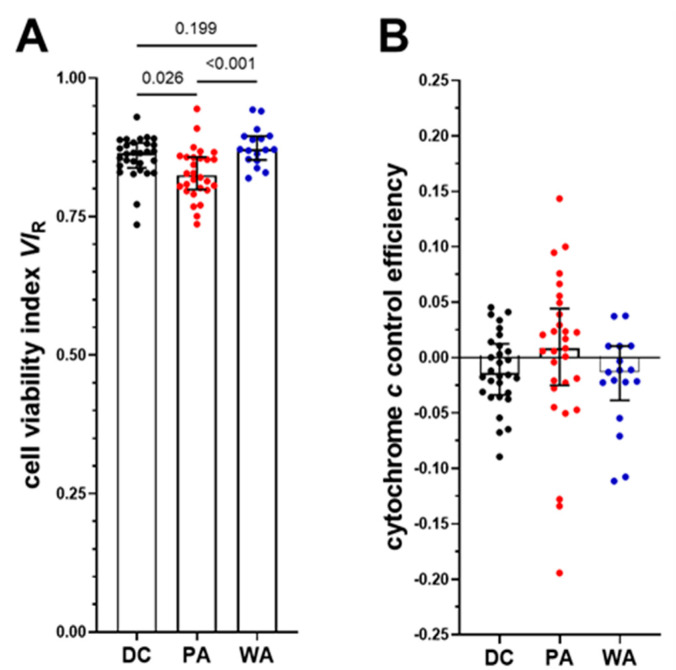
Effect of isolation methods and media on PLT viability. (**A**) Respirometric cell viability index (*VI*_R_) of PLT isolated by DC (black symbols), PA (red symbols), and WA (blue symbols). The respirometric cell viability index is *VI*_R_ = 1 − (*J*_ce5S_ − *J*_ce4Rot_)/(*J*_1Dig_ − *J*_2Ama_) [34]. (**B**) Cytochrome *c* control efficiency expresses the integrity of the outer mitochondrial membrane and is calculated as a change in respiration after the addition of cytochrome *c*, *j*_c_ = (*J*_1c_ − *J*_1Dig_)/(*J*_1c_ − *J*_2Ama_) [3,34]. Titration steps (ce4Rot, ce5S, 1Dig, and 1c) are depicted in Figure 2. One-way ANOVA with mixed-effects model, *p* values are shown above their respective pairwise comparison bar.

**Figure 6 biomedicines-09-01859-f006:**
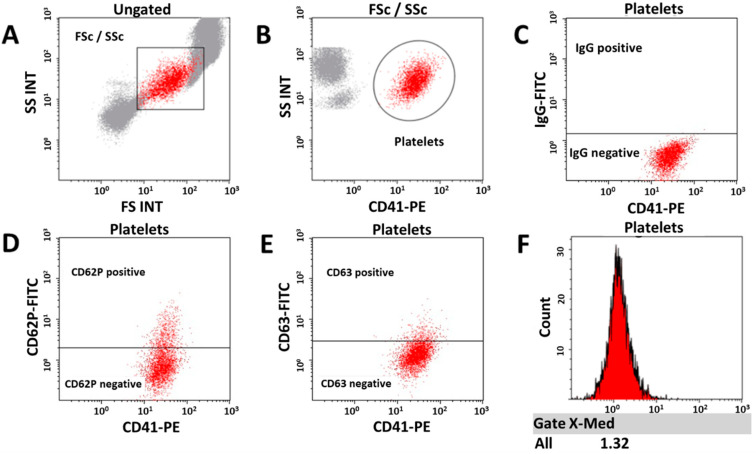
Expression of PLT activation markers. (**A**) Preliminary PLT gating based on morphological properties—forward scatter (FS) and side scatter (SS). (**B**) Precise PLT gating based on CD41 expression. (**C**) Positive threshold setting based on isotype control (IgG) expression. (**D**) Evaluation of CD62P activation marker expression. (**E**) Evaluation of CD63 activation marker expression. (**F**) Histogram to determine the median MFI of the marker of interest. This is the representative protocol for EDTA control blood samples.

**Figure 7 biomedicines-09-01859-f007:**
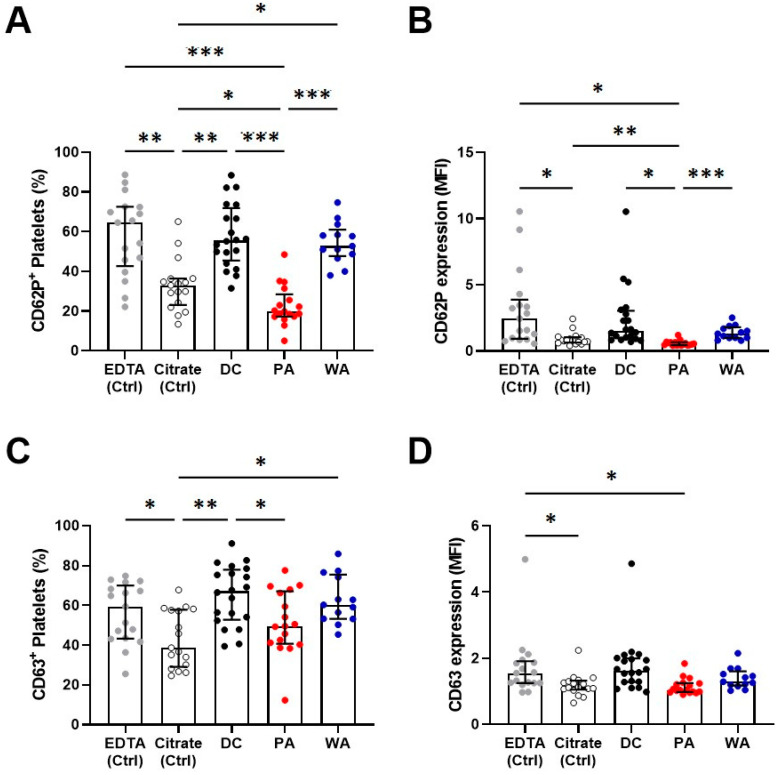
PLT activation in different isolation methods. (**A**) Percentage of PLT positive for CD62P. (**B**) Mean fluorescence intensity for CD62P. (**C**) Percentage of PLT positive for CD63. (**D**) Mean fluorescence intensity for CD62P. Ctrl, control blood samples before isolation procedures. 2-way ANOVA using multiple comparisons, * *p* < 0.05, ** *p* < 0.01, *** *p* < 0.001. Individual *p* values are shown in Appendix A.

**Figure 8 biomedicines-09-01859-f008:**
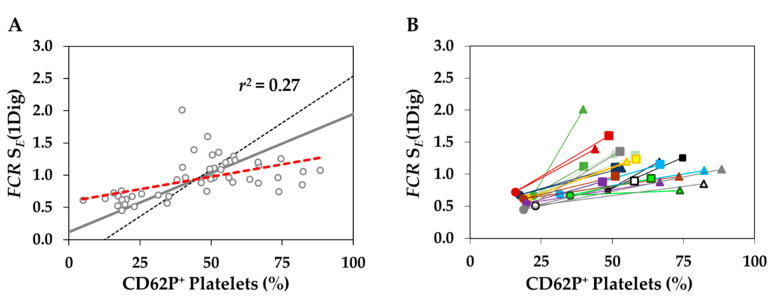
The relationship between PLT activation and mitochondrial respiration of PLT in the S-pathway. Respiration of permeabilized PLT in the S-pathway (S*_E_*, expressed as *FCR* normalized for ET capacity *E* of living cells, *FCR* = S*_E_*/*E*), and PLT activation expressed as a percentage of CD62P positive cells. (**A**) Regression analysis according to [49] where the red dotted line represents ordinary *Y/X* regression and the black dotted line represents ordinate projection of abscissal regression with inverted parameters *X/Y*, and the grey line represents mean regression line; *r*^2^ is independent of axes inversion. (**B**) Data points for individual blood donors are connected by a full line from DC (triangles) to PA (circles) and WA (squares); each color indicates a different donor.

## Data Availability

Original files are available Open Access at Zenodo repository: 10.5281/zenodo.5482452.

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
