# Peer review of "Mitochondrial Respiration of Platelets: Comparison of Isolation Methods"

_biomedicines, 2021, doi:10.3390/biomedicines9121859_

Round 1

Reviewer 1 Report

  1. Two multicomponent blood collection systems were used for platelet apheresis. Are there any differences between the results obtained with the two systems?
  2. Figure 1 should be refined. My suggestions: 1/ The volume of components added is not indicated for all tubes. 2/ The fraction used from the previous tube should be uniformly marked below the arrows and the centrifugation conditions above the arrows.
  3. In Figure 7, what does the CFA mean?
  4. Line 317, “PLT were not activated during DC and PA blood isolation (Figure 7).” This statement is true when comparing the results to the controls, however, the controls show significant activation, suggesting sample handling problems. How can this problem be explained?
  5. References:
  • The reference list is incomplete. The authors should refer to the latest paper by Siewiera K et al., (Sample Preparation as a Critical Aspect of Blood Platelet Mitochondrial Respiration Measurements–The Impact of Platelet Activation on Mitochondrial Respiration. J. Mol. Sci.202122(17), 9332; https://doi.org/10.3390/ijms22179332) which is closely related to the subject of the paper.
  • Lines 71-73: “However, different types of anticoagulants, centrifugation steps, and respiratory protocols affect the behaviour of PLT concentrates and the rate of mitochondrial respiration [27-31].” References 27, 29 and 30 are not appropriates as they do not deal with platelet respiration.

Author Response

Answers to Reviewer 2:

1. Two multicomponent blood collection systems were used for platelet apheresis. Are there any differences between the results obtained with the two systems?

Answer: Thank you for this question. Trima Accel MCS+ was used for patients No. 5, 7, 9, 11, 15, and 28 and Haemonetics MCS+ was used for the rest of our participants. We did not observe any significant effect nor trend in data obtained by these two systems and therefore we decided not to include this analysis in the manuscript (data not shown). This information was added into the text and readers, if interested, can download our data from the data repository to use it in further analysis. Since these results wouldn't be paired with each other as it was in case of PA and DC comparison and since they did not differ in any significant way, we decided not to include figures of these comparisons in the manuscript. It was not our goal to compare these two systems and this comparison would extend the frame of this manuscript, because different experimental protocols would be necessary. For our purpose of comparing DC and PA we used the isolation system which was at the moment accessible for us with regard to the current operations and workload of the transfusion department of our hospital.

We added into the manuscript:

Method section:

The Trima Accel MCS+ was used for patients No. 5, 7, 9, 11, 15, and 28 and Haemonetics MCS+ was used for the rest of our participants.

Discussion section:

…. samples and even though we used two different systems for PA, we did not observe any significant differences or trends between PLT isolated by (data not shown).

2. Figure 1 should be refined. My suggestions: 1/ The volume of components added is not indicated for all tubes. 2/ The fraction used from the previous tube should be uniformly marked below the arrows and the centrifugation conditions above the arrows.

Answer: Thank you very much for these constructive suggestions. We corrected Figure 1 accordingly.

3. In Figure 7, what does the CFA mean?

Answer: Thank you for pointing out our mistake. CFA was an abbreviation we used in our previous drafts of this manuscript. We corrected Figure 7 in concordance with the rest of the text.

4. Line 317, “PLT were not activated during DC and PA blood isolation (Figure 7).” This statement is true when comparing the results to the controls, however, the controls show significant activation, suggesting sample handling problems. How can this problem be explained?

Answer: Thank you for this excellent comment. Platelets are quite sensitive and fragile, and every intervention triggers some degree of their activation, such as the blood withdrawal itself (incl. diameter of the needle), their transfer as they hit the walls of the test tube, the ambient temperature changes etc. Citrate coated tubes are considered as a standard to determine platelet phenotype because it was previously shown that EDTA activates platelets (Holme et al. 1997, Cezary et al. 1998 - both references were added into the manuscript). So it was not surprising for us that markers of activation of EDTA platelets ​​are higher than for citrate platelets. In general, we would say that some degree of spontaneous activation is always related to platelet collection, and EDTA also has a known effect increasing their activation and important here is the change of these activation markers not between DC and PA, but regarding their respective controls EDTA and Citrate respectively. We added a short clarification in the discussion section in the manuscript including new references.

We changed a paragraph in the manuscript:

Discussion section:

Addressing PLT activation, we performed flow cytometry and assessed the panel of activation markers. Based on preliminary experiments we measured two membrane antigens CD62P and CD63. According to our results, PLT isolated by DC and WA showed higher activation in comparison to the PA group (Figure 7). It was reported previously, however, that some degree of spontaneous activation is always related to platelet collection and EDTA causes higher expression of activation markers in comparison to Citrate [47,48]. Taking into consideration that blood for each group (DC and PA) was withdrawn into tubes using different anticoagulant chemicals, EDTA and Citrate respectively, the important information is the change of activation markers in comparison to their respective controls. According to our results, neither DC nor WA caused higher PLT activation in comparison to their EDTA control. Similar trend was observed for PA, which did not exert higher activation than its control counterpart (Figure 7). Therefore, we conclude that the isolation process itself did not cause any additional activation of PLT.

5. The reference list is incomplete. The authors should refer to the latest paper by Siewiera K et al., (Sample Preparation as a Critical Aspect of Blood Platelet Mitochondrial Respiration Measurements–The Impact of Platelet Activation on Mitochondrial Respiration. J. Mol. Sci.2021, 22(17), 9332; https://doi.org/10.3390/ijms22179332) which is closely related to the subject of the paper.

Answer: We added this reference into our manuscript.

6. Lines 71-73: “However, different types of anticoagulants, centrifugation steps, and respiratory protocols affect the behaviour of PLT concentrates and the rate of mitochondrial respiration [27-31].” References 27, 29 and 30 are not appropriate as they do not deal with platelet respiration.

Answer: Reference numbers were reviewed and corrected.

Reviewer 2 Report

The manuscript is interesting. However, there are several question in the manuscript:

  1. The supplementary  figures and  results are lost. The authors should add these results in the manuscript.
  2. The authors should provide the function of platelets, such as the level of aggregation, not only surface marker.
  3. The authors should provide full text for all abbreviation
  4.  The authors should provide the advantage and disadvantage of this manuscript and others.
  5. The authors also should provide the advance or present study.
  6. The authors should provide the results about inhibitor of mitochondiral activation on platelet aggregation.
  7. The authors should provide the number of Institutional Review Board and research limitation in the manuscript.

Author Response

Answers to Reviewer 3:

1. The supplementary figures and results are lost. The authors should add these results in the manuscript.

Answer: Thank you for pointing this out. Accidentally, the file with supplementary data was not uploaded probably due to a technical mistake which we must have missed. We already sent these results to Biomedicines to add it to our final manuscript submission and you should see it online.

2. The authors should provide the function of platelets, such as the level of aggregation, not only surface marker.

Answer for comments 2, 5, 6, 7: Thank you very much for commenting on the aggregation capability of platelets. Unfortunately, we are not able to provide results on platelet aggregation, since we do not have fresh samples from our participants anymore and data from other blood donors would not be possible to pair with results presented in the manuscript. The platelet aggregometry was not included in our methods, because the primary focus of our manuscript was on non-aggregatory (i.e., non-hemostatic) functions and our goal was to describe and create a dataset of physiological respiration of human platelets isolated from the blood of healthy individuals, which could be later used as a baseline or controls for various pathologies.

We understand only platelet surface markers are not telling the whole story of platelets in hemostasis and we fully acknowledge the lack of platelet aggregometry method as one of the limitations of our study. Originally, we planned to include such measurements, however we ran into few methodological and ethical obstacles. It would be necessary to develop a protocol for platelet aggregometry in our hematology lab, which was not available for us at the time our project started. Since our samples of isolated platelets were suspended in various media (apheresis media and DPBS) it would not be comparable with standard control samples due to differences in light dispersion of these solutions. Also, to properly investigate aggregation of platelets (i.e. using various activators and inhibitors of aggregation etc.) acquired from platelet concentrates, we would need large volumes of blood to get sufficient platelet numbers in these concentrates. This would mean that the amount of whole blood drawn from our participants would significantly exceed the amount of blood which was approved by the Ethical committee due to the paired concept of our experimental groups (i.e., using platelets from one blood donor). Since our study was focused more on the non-aggregatory functions of platelets (in our case mitochondrial respiration) and considering reasons mentioned above, we decided to retract from platelet aggregometry in this paper. Second disadvantage and limitation of our experimental design is that based on our data we are not able to address the question of causality of platelet respiration and their activation.

To conclude, we think that the role of mitochondria during platelet aggregation deserves more detailed investigation which extends the frame of this manuscript and could be the advance of our project in future studies investigating the metabolism of platelets during hemostasis.

We added into the manuscript new figure (Figure 8) and following text:

Discussion section:

Lastly, we investigated the relationship between markers of PLT activation and mitochondrial respiration (Figure 8) by taking together data from all groups and performing regression analysis [49]. We did not find an apparent pattern, since all r2 values were quite low (data not shown). The highest correlation was found for the S pathway, when expressed as a relative FCR (r2 = 0.266; Figure 8A) suggesting a possible trend, especially when we paired results of PLT from different donors (Figure 8B).

Noteworthy, PLT isolated by PA from each donor were less activated and had lower respiration in S-pathway. Our results, however, do not demonstrate causality between PLT activation and differences in mitochondrial respiration, since we do not provide any data regarding negative and positive controls. This is one of the limitations of our study and to further clarify this phenomenon, more experiments using specific PLT activators and inhibitors are necessary. Second limitation of this study is that we do not provide data on PLT aggregatory functions. Considering that PLT activation to certain level is still a reversible process, specific measurements must be performed (such as optical aggregometry). In this study we demonstrated, that HRR is a method sensitive enough to study PLT metabolism, but to further investigate PLT metabolism during hemostasis exceeds the framework of this manuscript. Advantage of this study is that we provide a respirometry reference values of human PLT isolated by protocol acknowledged by international project MitoEAGLE [5] allowing our results to be used in comparative mitochondrial physiology in future projects.

3. The authors should provide full text for all abbreviation.

Answer: We made appropriate corrections and full text for all abbreviations was added according to the Instructions for authors.

4. The authors should provide the advantage and disadvantage of this manuscript and others.

Answer: The advantages of this manuscript and our experimental design:

1) Internationally acknowledged standardized methodology used for platelet isolation and high resolution respirometry measurements allowing our results to be used as a reference values in comparative mitochondrial physiology and possible application in future studies oriented both methodologically and clinically

2) Our experimental approach of platelets isolated from the same blood donors increases the relevance of our data and allowed us to performed regression analysis of respiration and activation for all groups together.

4) Showing high resolution respirometry as a sensitive method with possible applications in a clinical diagnostic using human platelets.

Disadvantages were elaborated in the answer above. Changes were made to text.

5. The authors also should provide the advance or present study.

Answer: please see the answer above.

6. The authors should provide the results about inhibitor of mitochondiral activation on platelet aggregation.

Answer: please see the answer above.

7. The authors should provide the number of Institutional Review Board and research limitation in the manuscript.

Answer: The Institutional Review Board was added to the manuscript. For research limitations of this manuscript please see answer above.

Round 2

Reviewer 1 Report

No further comments.

Reviewer 2 Report

The manuscript is good enough to publish.